# The Characteristics of Raindrop Size Distributions in Different Climatological Regions in South Korea

**Cheol-Hwan You** [1], **Hyeon-Joon Kim** [2], **Sung-Ho Suh** [3], **Woonseon Jung** [4] and **Mi-Young Kang** [1,*]

1   Atmospheric Environmental Research Institute (AERI), Pukyong National University, Busan 48513, Korea
2   Department of Civil and Environmental Engineering, Chuang-Ang University, Seoul 06974, Korea
3   Flight Safety Technology Division, NARO Space Center, Korea Aerospace Research Institute (KARI), Goheung-gun 59571, Korea
4   National Institute of Meteorological Sciences, 33 Seohobuk-ro, Seongwipo-si 63568, Korea
*   Correspondence: miyoungkang@pknu.ac.kr; Tel.: +82-51-629-6639

**Abstract:** To understand the microphysical characteristics of rainfall in four different climatological regions (called BOS, BUS, CPO, and JIN) in South Korea, DSDs and their variables, including the mass-weighted mean diameter ($Dm$) and normalized number concentration ($logNw$), were examined. To examine the characteristics of DSDs at four sites with different climatology and topography, data measured from Parsivel disdrometer and wind direction from Automatic Weather System (AWS) during rainy seasons from June to August for three years (2018 to 2020) were analyzed. The DSDs variables were calculated using Gamma distribution model. In the coastal area, larger raindrops with a lower number concentration occurred, whereas smaller raindrops with a higher number concentration dominated in the middle land and mountain region. The mountain area of CPO and middle land area of JIN had a larger contribution to the rain rate than that of the coastal area of BOS and JIN in the range of the smallest diameter. The contribution of the drop size to the total number concentration at the CPO and JIN sites was larger (smaller) than that at BOS and BUS in the smallest (larger) diameter. The average shape and slope parameter of gamma model were higher values at the mountain area than at other sites for both rain types, Z-R relation and polarimetric variables were also shown different values at the four studied sites. The intercept coefficient of Z-R relation showed higher values in the mountain area and middle land area than the coastal area. The slope values of Z-R relation were the smallest in the mountain area. The polarimetric variables of $Z_H$ and $Z_{DR}$ were shown highest (lowest) value at the coastal region of BOS (mountain area of CPO) site for both rain types. The $Dm$-rose, which shows the $Dm$ distributions with the wind direction, was used in this study. In the coastal area (mountain and middle land area), the dominant wind was east–southeast (east) direction. The ratio of the smaller diameter to the middle size at BOS was much smaller than that at CPO. In the analysis of the hourly distribution of the Dm and $logNw$, there were two and four peaks of Dm at BUS and BOS, respectively. There was one peak of the Dm at the CPO and JIN sites. The time variation of the Dm was much higher than that of the $logNw$.

**Keywords:** drop size distributions; continental and coastal area; Dm-rose; rain rate; number concentration

## 1. Introduction

Raindrop size distributions (DSDs) provide essential information about the microphysical characteristics of precipitation. DSDs also play a fundamental role in obtaining accurate rainfall estimations from weather radars and radio frequency conduction in the atmosphere [1,2].

There have been many DSD studies on its modeling based on observation, fitting the distribution with power-law [3], gamma distribution [4], and normalized gamma distribution [5]. To understand DSDs with rainfall types, many previous studies have focused on classifying rainfall into convective and stratiform types [6–8]. They have

proposed a rainfall classification method using DSD parameters, such as the intercept, median diameter, normalized number concentration, and rain rate.

The spatial distributions of DSDs were retrieved from polarimetric radars to understand the microphysical characteristics and to improve the performance of radar rainfall estimation using a constrained gamma DSD model [9], a Bayesian approach [10], and a gamma DSD model [11]. Maki et al. [12] examined the effects of natural variations in rain DSDs on the rain rate estimates of an X-band polarimetric radar. The rainfall estimator $R(K_{DP}, Z_{DR})$ is the least sensitive to variations in DSDs, and $R(Z_H)$ is the most sensitive to them. Zhang et al. [13] improved the parameterization of rain microphysics obtained from the disdrometer and polarimetric radar for a numerical model using Kessler-type parameterization. They found that simplified constrained-gamma DSD model parameterization outperformed Marshall–Parmel model parameterization.

Power-law and gamma DSDs models have been used to describe the microphysics of widespread precipitation. Smith and Kliche [14] simulated power-law and gamma DSD models to understand their biases in terms of moment methods and found that the bias was stronger when high-order moments were used for the calculation. Montopoli et al. [15] proposed a new stochastic vector autoregressive semi-Markov model to generate the time series of the three driving parameters of a normalized gamma DSD using a Joss–Waldvogel disdrometer in the UK.

There have been also many studies on the characteristics of DSDs in different precipitation types, including thunderstorms [16,17], heavy rainfall [18], and squall-line events [11]. The characteristics of DSDs with rainfall types were examined in Italy [19] and south India [20]. In order to reduce the number of parameters of the gamma DSD model and to improve the radar rainfall estimation, the shape ($\mu$) and slope ($\Lambda$) relations have been examined in many studies [21–24].

To understand the spatial variability of DSDs, Jaffrain et al. [25] examined the variability of DSDs using 16 Parsivel disdrometers spaced every $1 \times 1$ km$^2$ in Switzerland. They found that the significant variability of the DSDs at a small scale was in the order of 20% for the total number concentration ($N_t$), 10% for the mass-weighted diameter ($D_m$), and 30% for the rain rate. Bringi et al. [26] estimated the spatial correlation between the DSD variables and the rain rate for the two precipitation types using S-band polarimetric radar and a 2-D video disdrometer. They found that the convective rain showed a shorter decorrelation distance than stratiform rain in the rain rate, and the correlation between the DSD variables and the rain rate from the radar and 2-D video disdrometer were in good agreement over distances ranging from 1.5 to 7 km.

Leinonen et al. [27] investigated the climatological characteristics of DSDs for five years in Finland to improve the understanding of high-latitude rain microphysics and to provide implications for ground-based and spaceborne radars. Bringi et al. [28] investigated the characteristics of DSDs using a normalized Gamma model in different climatic regions. They classified the precipitation into two rain types, including convective and stratiform rain, and analyzed the gamma parameters using DSDs observed (retrieved) using S-band polarimetric radar.

The regional variability of DSDs has been examined in many different countries. Marzuki et al. [29] examined the regional variability of raindrop size distribution using Parsivel disdrometer data at four different sites at a similar latitude across Indonesia. They found that regional variability is closely related to the variation of the terrain, mesoscale convective system propagation, and the horizontal landmass scale. Lam et al. [30] investigated the characteristics of DSDs in equatorial Malaysia using three years' worth of JWD data. They compared Gamma distribution and normalized Gamma distribution derived from the moment method and maximum likelihood estimation (MLE) and found the normalized Gamma model with MLE provided better accuracy for long-term rain rate statistics. Malinga and Owolawi [31] investigated various parameters using Gamma and normalized Gamma models obtained using JWD data in South Africa and derived the relations between the raindrops and rain rate in many different rain types. Kozu et al. [32]

examined seasonal and diurnal variations of DSDs in three different areas, Gandanki, Singapore, and Kototabang, using the moment method with three, four, and six moments. Montopoli et al. [33] analyzed the characteristics of DSD parameters for convective and stratiform rain in terms of a normalized gamma model in four different climatic regimes, the UK, Greece, Japan, and the US.

Seela et al. [34] examined the raindrop size distribution characteristics of summer and winter seasons in north Taiwan using 12 years DSD data measured from Joss-Waldvogel disdrometer (JWD). They found that winter rainfall has a higher number concentration of small raindrops and a lower number concentration of midsize and larger drops than those summer rainfall. They also found that the average Dm is higher in convective rainfall than stratiform rainfall for both seasons. Oue et al. [35] investigated the two types of precipitation particle distribution in convective cells accompanying a Baiu frontal rainband around Okinawa Island in Japan using C-band polarimetric radar and JWD. They compared DSD and radar variables for convective cells and stratiform cells. They found that higher (lower) number densities of small raindrops of 1–2 mm were dominated in stratiform (convective) cells. Chen et al. [36] analyzed the raindrop size distribution in the Meiyu season in Eastern China with rain types. They found that the mean *logNw* and *Dm* values are 3.8 (3.45) and 1.71 (1.3) mm for convective (stratiform) rain.

In the research on the decadal variability of DSDs in Busan, Korea, a higher rain rate, greater annual average Dm, and higher frequency of convective rain were shown in 2012 compared to 2002 [37]. The diurnal DSD variability, which shows maritime (continental) rainfall occurring in the daytime (night time), was examined using wind direction analysis in a coastal area in Korea [38]. Recently, Cha and Yum [39] examined the characteristics of DSDs in mountain and coastal regions using 2 years' worth of Parsivel disdrometer data in Korea. They focused on the analysis of DSDs with different temperatures and found that the relatively warm temperature ($-5$ to $0\,^\circ$C) increased the snow particle number densities at around 0.6–1 mm diameter and the relatively cold temperature ($-15$ to $-10\,^\circ$C) decreased it above 2 mm diameter.

There are still few studies on DSD parameters in the different climatological regions and especially with wind direction in Korea. To understand the characteristics of DSDs, Z-R relation, and polarimetric variables in different climatological regions in Korea, the observed DSDs from Parsivel at four different sites (two sites in a coastal area and lower altitude, one site in a middle land area, and one site in a mountain area) were examined. A new way of showing the averaged diameter with wind direction called Dm-rose was also proposed. In Section 2, the data and adopted methodology are described. The obtained results are explained in Section 3 and summarized in Section 4.

## 2. Materials and Methods

### 2.1. Datasets

To examine the characteristics of DSDs at 4 sites with different climatology and topography in South Korea, data obtained from PARSIVEL disdrometers during the rainy seasons from June to August for three years (2018 to 2020) were used. Figure 1 shows the locations of the four studied sites, including Daegwallyeong (hereafter CPO, 37.68709°N, 128.7587°E), Jincheon (hereafter JIN, 36.98°N, 127.44°E), Boseong (hereafter BOS, 34.76335°N, 127.2123°E), and Busan (hereafter BUS, 35.12°N, 129.1°E). CPO (JIN) is located at a relatively high (middle) latitude and higher (lower) altitude, whereas both BOS and BUS are located at a relatively lower latitude and coastal area. These locational differences among sites have resulted in distinguishable precipitation climatology. The yearly average rainfalls (10-year trend) for the last 30 years from 1991 to 2020 are around 1695.1 mm ($-338.5$ mm) for CPO, 1218.7 mm ($-26.7$ mm) for JIN, 1449.2 mm ($+92.8$ mm) for BOS, and 1576.7 mm ($+32.4$ mm) for BUS. Cheonan is the nearest official station to JIN, Goheung is the nearest site to BOS, and Pusan is the nearest site to BUS. Figure 2 shows the time series of annual rainfall for 30 years and the 10-year trend at each site.

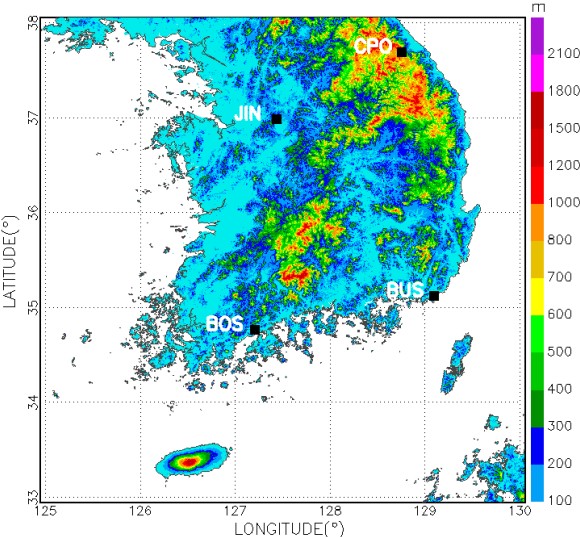

**Figure 1.** The locations of the four PARSIVEL disdrometers (full circle), with topography. CPO site is located at the mountain area (849.15 m in altitude), JIN is a middle land area (111 m in altitude), BUS (1.41 m in altitude) and BOS (10 m in altitude) in a coastal area.

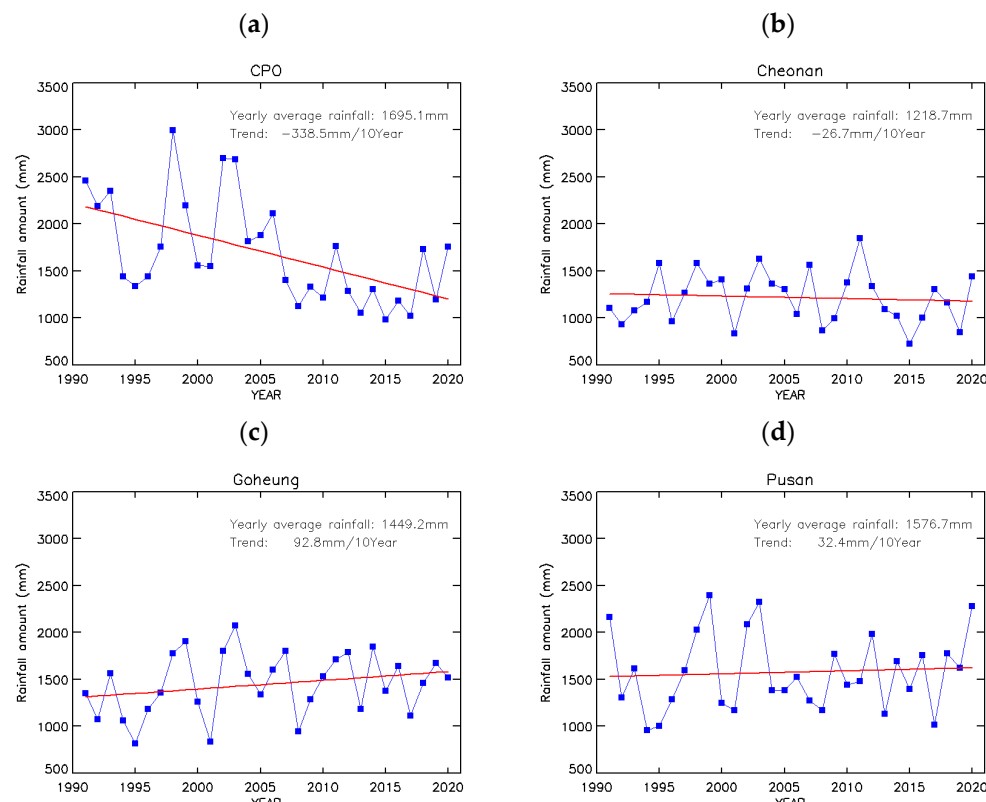

**Figure 2.** The time series of annual rainfall (blue) and the trend (red) for 30 years at (**a**) CPO, (**b**) JIN, (**c**) BOS, and (**d**) BUS site. Cheonan is the nearest official station to JIN, Goheung is the nearest site to BOS, and Pusan is the nearest site to BUS.

The Parsivel disdrometer is a laser-optic system that measures 32 diameter channels from 0.062 mm to 24.5 mm; its detailed specifications are shown in [40]. The first two channels among 32 bins were not included in the analysis because of the low SNR (Signal-to-Noise Ratio) [41,42]. The Parsivel diserometer is widely used for the study on DSDs analysis due to its easy installation and cheap expense. However, there are four erroneous factors such as strong wind effect, marginal faller, and splashing effect. To remove these

errors, the quality control algorithm for the Parsivel disdrometer was applied using the scheme proposed by Friedrich et al. [41].

$$\left| V_{obs} - V_{ref} \right| < 0.6V_{ref} \tag{1}$$

where $V_{obs}$ is fall velocity observed by Parsivel disdrometer and $V_{ref}$ is the reference fall velocity proposed by [43].

$$V_{ref} = 9.65 - 10.3exp(-0.6D) \tag{2}$$

The raindrop spectra were selected for the analysis when the observed fall velocity is satisfied with Equation (1).

In addition to this algorithm, if the 1 min rain rate was less than 0.1 mm h$^{-1}$ or higher than 200 mm h$^{-1}$, the corresponding spectra were removed as unreliable data. The sample numbers used for the analysis after the quality control algorithm for each site were 30,605 for BOS, 34,890 for BUS, 49,818 for CPO, and 28,041 for JIN. To propose a way of showing the *Dm* distribution with wind direction, the wind direction measured by Automatic Weather System (AWS) as the same period of Parsivel disdrometer.

### 2.2. Methodology

After the data quality control algorithm was applied to the DSD data, the DSD parameters were calculated using the Gamma distribution model proposed by Ulbrich [4], as shown in Equation (3).

$$N(D) = N_0 D^\mu \exp(-\Lambda D) \tag{3}$$

where $N(D)$ (mm$^{-1}$ m$^{-3}$) is the number concentration per unit volume for each drop diameter, $N_0$ (mm$^{-1-\mu}$ m$^{-3}$) is an intercept parameter, $\Lambda$ (mm$^{-1}$) is the slope, and $\mu$ (dimensionless) indicates the shape parameter of the Gamma model. The n$_{th}$ moment for the DSDs, $M_n$, was calculated using Equation (4). Moments of the 2nd, 4th, and 6th orders were used to describe the DSD parameters in this study.

$$M_n = N_0 \Lambda^{-(\mu+n+1)} \Gamma(\mu+n+1) \tag{4}$$

Then, the Gamma DSD variables were calculated as follows [23]:

$$\mu = \frac{(7-11\eta) - \left(\eta^2 + 14\eta + 1\right)^{1/2}}{2(\eta-1)}, \tag{5}$$

$$\Lambda = \left[ \frac{M_2 \Gamma(\mu+5)}{M_4 \Gamma(\mu+3)} \right]^{1/2} = \left[ \frac{M_2(\mu+4)(\mu+3)}{M_4} \right]^{1/2}, \tag{6}$$

where $\eta$ is the ratio of moments ($\eta = M_4^2 / M_2 M_6$).

The additional DSD variables were calculated as follows [5] for the normalized gamma distributions:

$$D_m = \frac{M_4}{M_3}, \tag{7}$$

$$LWC = \frac{\pi}{6} \rho_w M_3, \tag{8}$$

$$N_w = \frac{4^4}{\pi \rho_w} \left( \frac{LWC}{D_m^4} \right), \tag{9}$$

where $D_m$ is the mass-weighted mean diameter (mm) and $N_w$ is the normalized intercept parameter (mm$^{-1}$m$^{-3}$).

To find the differences in the DSD characteristics among rainfall types, the classification scheme proposed by [28] was applied to the data. When the 1 min rainfall rate is higher than 5 mm h$^{-1}$ (0.5 mm h$^{-1}$) and the standard deviation for five minutes rainfall rate is

larger (smaller) than 1.5 mm h$^{-1}$, then it is classified as convective (stratiform) rain. They used a 2 min averaged rain rate for the analysis; however, 1 min data were used in this study to retain more sampling numbers.

To compare Z-R relation and polarimetric variables at each site, T-matrix scattering simulation was executed. The T-matrix scattering simulation derived by Waterman [44] and programmed by Leinonen [45] was used in this study. The raindrop shape was represented by the combined drop axis ratio proposed by Beard and Kubesh [46]'s model for raindrops between 0.7 and 1.5 mm, by Thurai and Bringi [47]'s model for raindrop diameters larger than 1.5 mm, and by sphere model for raindrop diameter smaller than 0.7 mm. Additional parameters in the scattering simulation are the distribution of raindrop canting angles, the temperature, and the wavelength. In this study, the temperature is set to 20 °C and the distribution of canting angles is assumed to be Gaussian with an average of 0° and a standard deviation angle of 10°. The wavelengths for the simulation are 10.39 cm for CPO, 10.97 cm for JIN, 10.38 cm for BOS, and 11.06 cm for BUS. The rain rate was calculated by the following equation:

$$R = \frac{\pi}{6} \int_{D_{min}}^{D_{max}} \rho_w V_t(D) N(D) dD, \qquad (10)$$

where $\rho_w$ is water density (g m$^{-3}$), $V_t$ (m s$^{-1}$) is the terminal velocity of raindrops with size. The relation between fall velocity and raindrop size is given by Atlas et al. [42].

To propose *Dm*-rose, the wind direction was selected when wind speed is stronger than 0.2 m s$^{-1}$ and divided into 16 directions. Then, average Dm values and their frequency with 16 wind directions were calculated. The workflow of this study is shown in Figure 3.

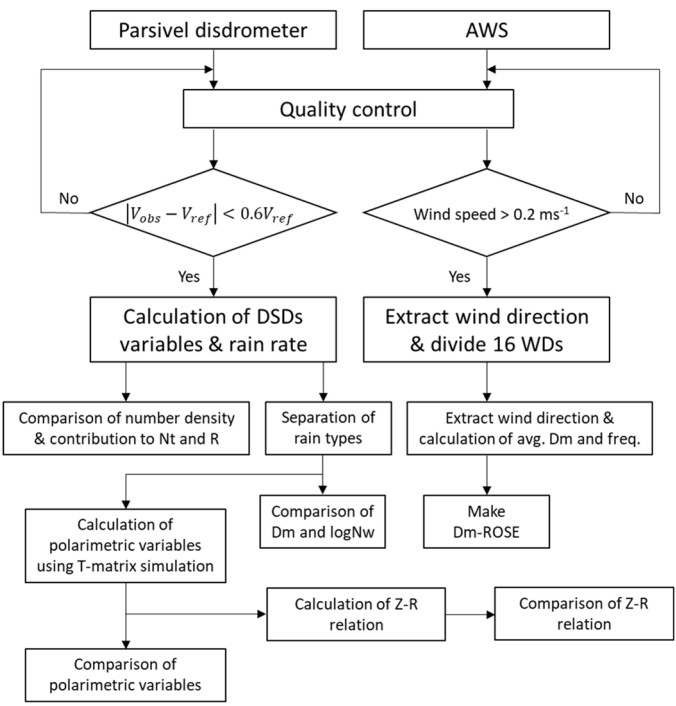

**Figure 3.** The workflow of this study.

## 3. Results

### 3.1. Average Drop Size Distributions

To understand the raindrop number concentration with the diameter at each site, average N(D)s were calculated (Figure 4). In the range smaller than 1 mm diameter, the CPO and JIN sites had larger number concentrations than the BUS and BOS sites. Between 1.5 mm and 4 mm in diameter, the number concentration of BOS (CPO) was larger (smaller) than at the other sites. In the range larger than 4 mm in diameter, BOS (BUS) had the largest (smallest) number density. The average *Dm* (*LogNw*) values of BOS, BUS, JIN, and CPO sites

were 1.30 mm (3.58 mm$^{-1}$m$^{-3}$), 1.15 mm (3.81 mm$^{-1}$m$^{-3}$), 1.10 mm (3.88 mm$^{-1}$m$^{-3}$) and 1.06 mm (4.06 mm$^{-1}$m$^{-3}$), respectively. At the BUS and BOS sites in the coastal area, the rainfall consisted of larger raindrops with a lower number concentration. In the mountain area of the CPO and the middle land area of JIN sites, smaller raindrops with a higher number concentration dominated.

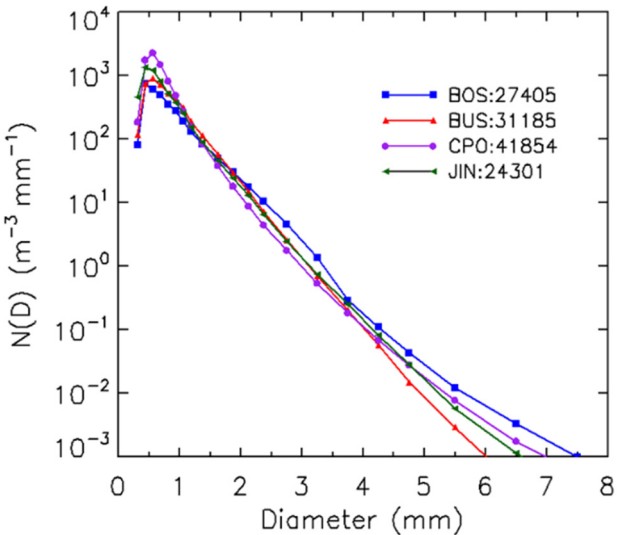

**Figure 4.** Average raindrop number concentration with diameter at four different sites. The rectangles in blue represent BOS, the triangles in red represent BUS, the circles in purple represent CPO, and the arrows in green represent the JIN site. The numbers in the legend are sampling numbers at each site.

To find the number concentration of four different sites at the same rain rate, the rain rates were categorized into the three ranges of 0.1 < R < 5.0 mm h$^{-1}$ (weak rain rate), 5.0 < R < 10.0 mm h$^{-1}$ (moderate rain rate), and 10.0 mm h$^{-1}$ < R (severe rain rate). Figure 5 shows the number concentration of the diameters with the three different rain rate categories. At the number concentration of a raindrop diameter of less than 1 mm, the CPO (BUS) site had the largest (smallest) number density in all rain rate categories. There were no samples observed at a diameter larger than 5 mm, and the three sites of BUS, CPO, and JIN had similar number concentrations in the weak rain rate category (Figure 5a). At the moderate rain rate (Figure 5b), the number density of the smaller diameter (less than 1 mm) at CPO was much higher than at the other sites, and the BOS site had a larger number concentration of middle and large size of diameters (larger than 1 mm). At the severe rain rate, the number density of all sites had similar patterns except for the BUS site, with a drop diameter of larger than 4 mm (Figure 5c).

To determine the contribution of the drop diameter to the rain rate and total number concentration, the drops were divided into four categories (D < 1 mm, 1 < D < 2 mm, 2 < D < 3 mm, and 4 mm < D). Figure 6 shows the contribution ratio of the rain rate to the total number concentration with respect to each raindrop diameter. In the case of the rain rate, its contribution to the raindrop diameter category of less than 1 mm was the largest (smallest) at the CPO (BOS) site. The rain rates at the CPO and JIN sites had larger contributions than at the BUS and BOS sites in the range of diameters of less than 1 mm. The raindrop category of 1 mm to 2 mm had the largest (smallest) contribution at the BUS (CPO) site. The mountain area of CPO and middle land area of JIN had a larger rain rate contribution than that of the coastal area of BOS and BUS in the range of the smallest diameter (less than 1 mm). For the larger raindrop diameter (larger than 1 mm), the pattern shows the reverse. More than 80% of the contributions of rain at all sites were less than 2 mm in diameter, whereas more than 79% of the contributions of the total number concentration occurred at a diameter of less than 1 mm. Similar to the rain rate contributions, the contributions of the total number concentration at CPO and JIN

had larger values than those at BOS and BUS in a range of diameter less than 1 mm. At a diameter larger than 1 mm, BOS and BUS had larger contributions than those at CPO and JIN.

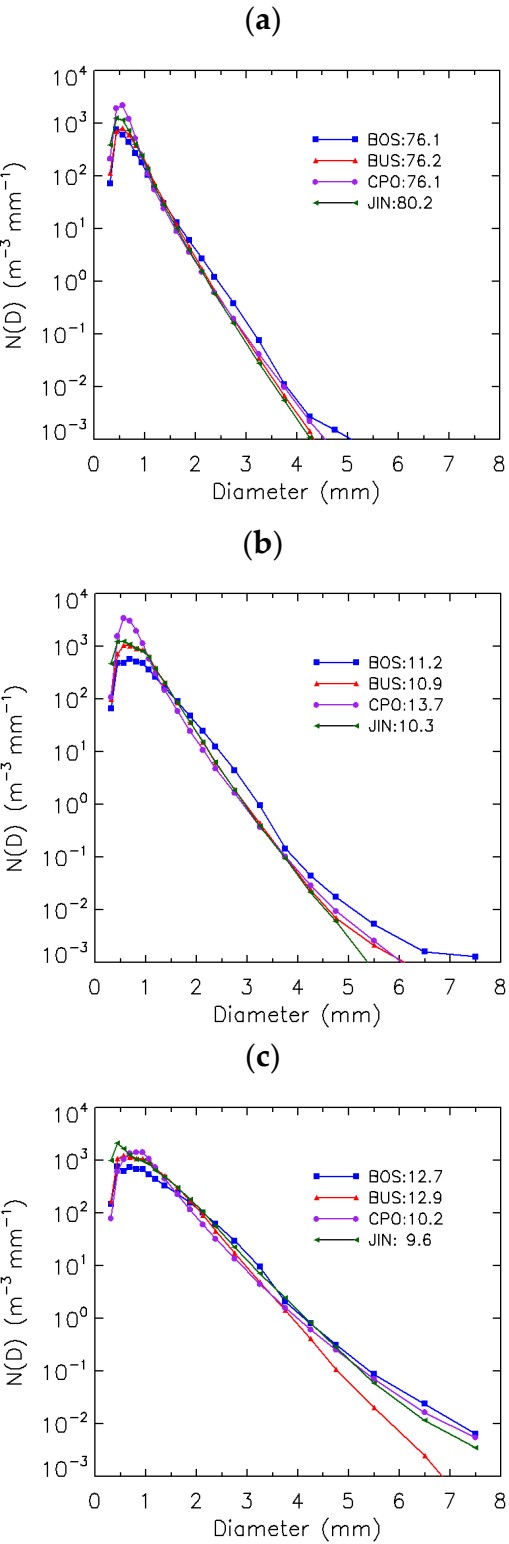

**Figure 5.** Average raindrop number concentration with diameter at four different sites with respect to rain rate (**a**) 0.1 < R < 5.0 mm h$^{-1}$, (**b**) 5.0 < R < 10.0 mm h$^{-1}$, and (**c**) 10.0 mm h$^{-1}$ < R. The rectangles in blue represent BOS, the triangles in red represent BUS, the circles in purple represent CPO, and the arrows in green represent the JIN site. The numbers in the legend represent the percentage of each rain rate.

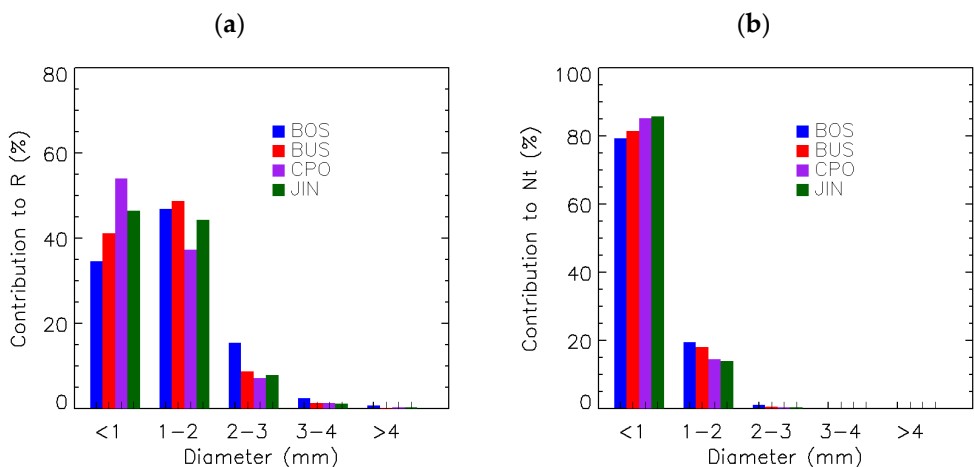

**Figure 6.** The contribution ratios of the raindrop diameter to (**a**) the rain rate and (**b**) the total number concentration. The blue represents BOS site, the red represents BUS site, the purple represents CPO site, and the green represents JIN site.

### 3.2. Average Drop Size Distributions with Rain Types

To determine the characteristics of DSDs with rain types, stratiform and convective rain were separated. Figure 7 shows the number concentrations of the diameters at four different sites with the rain types. The percentages of stratiform (convective) rain at BOS, BUS, CPO, and JIN were 66% (34%), 68% (32%), 71% (29%), and 75% (25%), respectively. The occurrence frequency of convective rain was higher in the coastal area of BOS and BUS than in the mountain area of CPO and the middle land area of JIN. The CPO site had the highest number concentration in both rain types at a raindrop diameter less than 1 mm. BOS had the highest number concentration at a diameter larger than 2 mm for stratiform rain. For convective rain, the BOS (BUS) site had the smallest number concentration at a raindrop size of less (larger) than 2 mm (4 mm) in diameter.

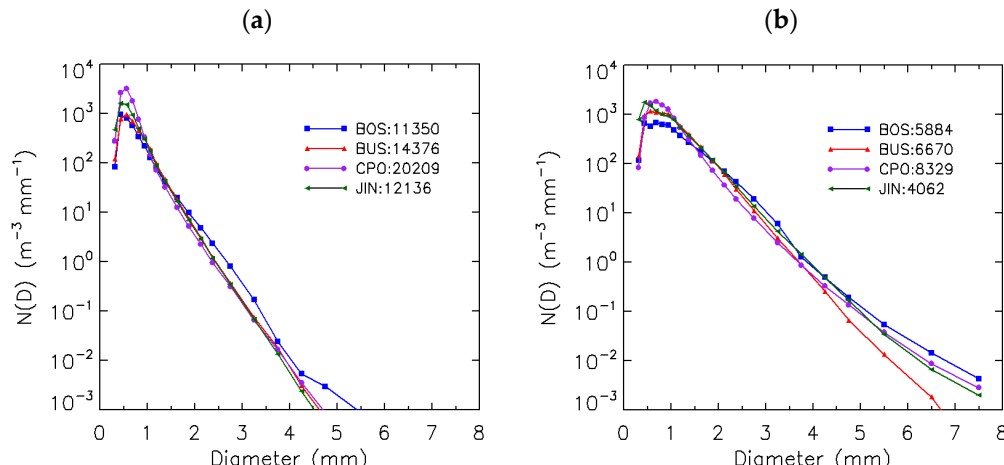

**Figure 7.** Average number concentrations of diameter with rain types (**a**) stratiform and (**b**) convective rain at the four studied sites. The number in the legend represent the sample number. The rectangles in blue represent BOS, the triangles in red represent BUS, the circles in purple represent CPO, and the arrows in green represent the JIN site.

The average $Dm$ values at BOS, BUS, CPO, and JIN for stratiform (convective) rain were 1.24 mm (1.84 mm), 1.11 mm (1.55 mm), 0.97 mm (1.45 mm), and 1.05 mm (1.59 mm), respectively. The average $logNw$ at BOS, BUS, CPO, and JIN for stratiform (convective) rain were 3.62 mm$^{-1}$m$^{-3}$ (3.70 mm$^{-1}$m$^{-3}$), 3.83 mm$^{-1}$m$^{-3}$ (4.04 mm$^{-1}$m$^{-3}$), 4.23 mm$^{-1}$m$^{-3}$ (4.14 mm$^{-1}$m$^{-3}$), and 3.99 mm$^{-1}$m$^{-3}$ (4.02 mm$^{-1}$m$^{-3}$), respectively. The mean rain rates at BOS, BUS, CPO, and JIN for stratiform (convective) rain were 1.96 mm h$^{-1}$ (17.97 mm h$^{-1}$),

1.98 mm h$^{-1}$ (17.31 mm h$^{-1}$), 2.30 mm h$^{-1}$ (13.93 mm h$^{-1}$), and 2.03 mm h$^{-1}$ (18.56 mm h$^{-1}$), respectively (Table 1).

**Table 1.** The average DSDs variables, rain rates, and sample numbers of rain types at the four studied sites.

| | BOS | | BUS | | CPO | | JIN | |
|---|---|---|---|---|---|---|---|---|
| Types | Str. | Con | Str. | Con. | Str. | Con. | Str. | Con. |
| $Dm$ (mm) | 1.24 | 1.84 | 1.11 | 1.55 | 0.97 | 1.46 | 1.05 | 1.59 |
| $LogNw$ (mm$^{-1}$ m$^{-3}$) | 3.62 | 3.70 | 3.83 | 4.04 | 4.23 | 4.14 | 3.97 | 4.02 |
| Rain rate (mm h$^{-1}$) | 1.96 | 17.97 | 1.98 | 17.31 | 2.30 | 13.93 | 2.03 | 18.56 |
| No. (%) | 11350 (66) | 5884 (34) | 14376 (68) | 6670 (32) | 20209 (71) | 8329 (29) | 12136 (75) | 4062 (25) |

Figure 8 shows the occurrence frequency of shape parameter with rain types at the four different sites. The frequency of convective rain was larger than that of stratiform rain less than 7 for BOS, 7 for BUS, 7 for JIN, and 13 for CPO. There is one peak value of the frequency in stratiform rain at the four studied sites, whereas there are two peaks in convective rain at the CPO site. The average value of shape parameters for convective (stratiform) rain at the BOS, BUS, CPO, and JIN sites were 4.75 (7.70), 5.70(9.61), 7.74 (13.25), and 5.56 (9.82), respectively.

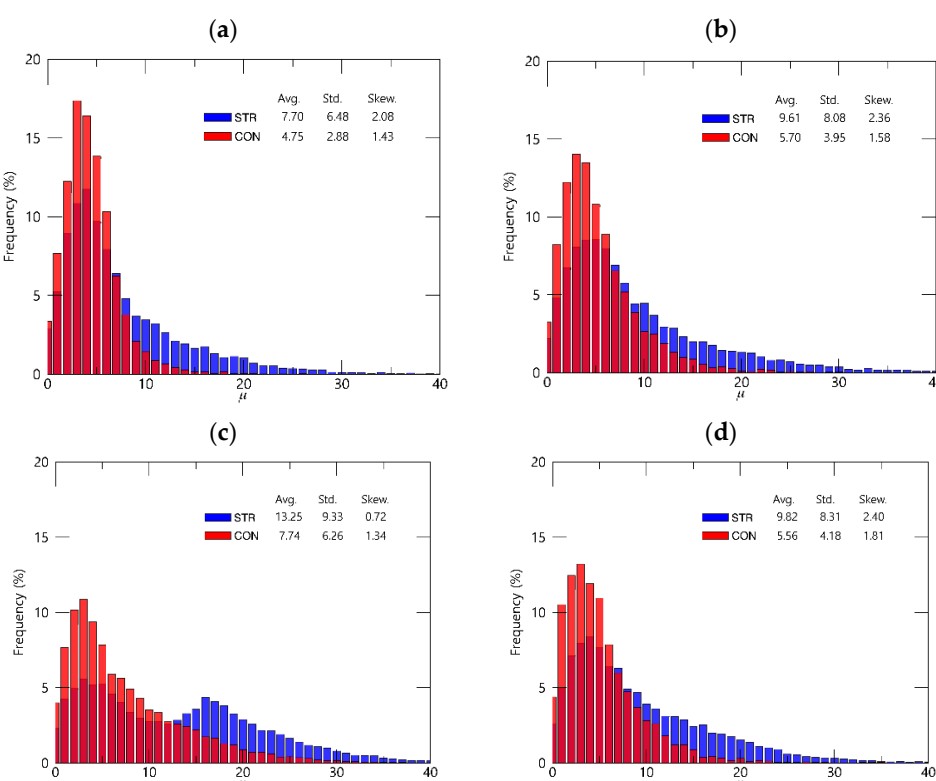

**Figure 8.** The occurrence frequency of shape parameters with rain type at (**a**) BOS, (**b**) BUS, (**c**) CPO, and (**d**) JIN site. The blue represents stratiform rain and the red represents convective rain. The legend shows the average (Avg.), standard deviation (Std.), and skewness (Skew) of shape parameter.

Figure 9 shows the occurrence frequency of slope parameter with rain types at the four different sites. The occurrence frequency of convective rain was larger than that of stratiform rain less than 7 mm$^{-1}$ at BOS, less than 9 mm$^{-1}$ at BUS, less than 9 mm$^{-1}$ at JIN, and less than 17 mm$^{-1}$ at CPO. There is also one peak value of the frequency in stratiform rain at the four studied sites, whereas there are two peaks in convective rain at CPO site. The average value of shape parameters for convective (stratiform) rain at the

BOS, BUS, CPO, and JIN sites were 5.18 mm$^{-1}$ (11.95), 6.99 mm$^{-1}$ (14.66), 10.08 mm$^{-1}$ (22.77), and 6.88 mm$^{-1}$ (16.13), respectively.

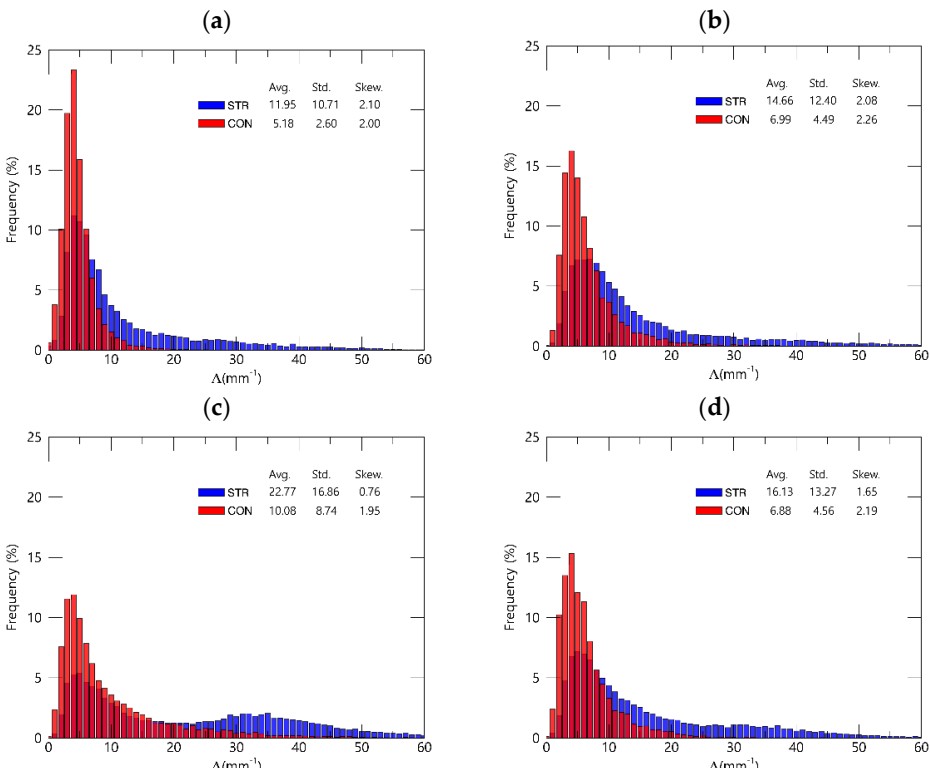

**Figure 9.** The occurrence frequency of slope parameters with rain type at (**a**) BOS, (**b**) BUS, (**c**) CPO, and (**d**) JIN site. The blue represents stratiform rain and the red represents convective rain. The legend shows the average (Avg.), standard deviation (Std.), and skewness (Skew) of slope parameter.

Figure 10 shows a boxplot of the *Dm* and *logNw* average values for the stratiform and convective rain at each site, with the separation line of the rain types proposed by Bringi et al. [8] (BR09). The average *Dm* and *logNw* values at the four studied sites almost coincide with the BR09 line. The number concentrations for the stratiform and convective rain are not very different among the sites, but the average diameter has larger differences between the rain types.

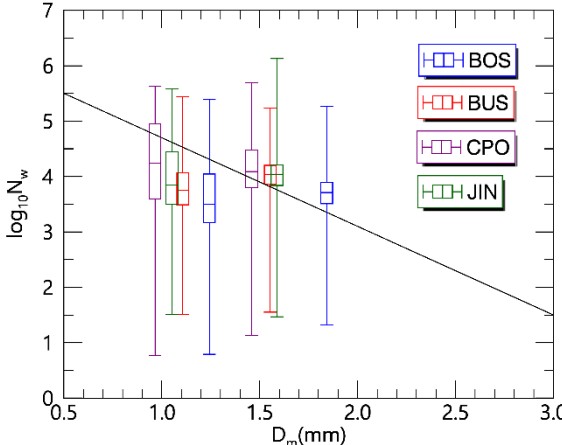

**Figure 10.** Boxplot of *Dm* and *logNw* values of convective and stratiform rain for each site. The line curve represents the separation line of rain types proposed by BR09. The section above (below) this line represents convective (stratiform) rain.

### 3.3. The Characteristics of Z-R Relationship and Polarimetric Variables

In order to calculate the Z-R relation (R(Z)), rain rate and reflectivity were calculated by 1 min DSD spectra. Figure 11 shows the scatter plot of rain rate measured by DSDs and obtained Z-R relations at the four studied sites. The intercept values of the relations at the BOS, BUS, and JIN sites were around 0.06 and 0.1 at the CPO site. The slope values of Z-R relations were 0.62 at BOS, 0.64 at BUS, 0.63 at JIN, and 0.57 at CPO. The correlation coefficients of the relations at the four studied sites were distributed between 0.94 and 0.96. The most accurate relation with 2.647 mm h$^{-1}$ of RMSE was obtained at the BUS site.

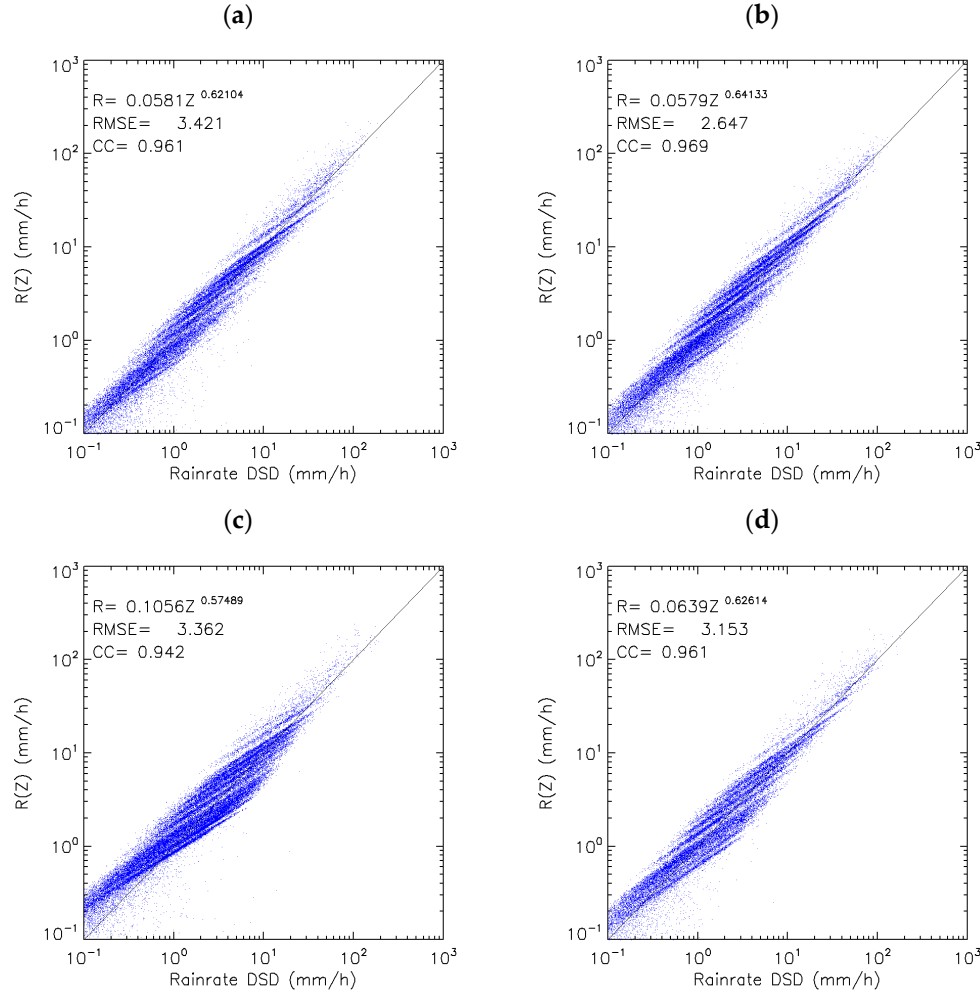

**Figure 11.** The scatter plots of rain rate measured by DSD and R(Z) relations were obtained from DSDs at (**a**) BOS, (**b**) BUS, (**c**) CPO, and (**d**) JIN sites. The legend shows the Z-R relation, root mean square error (RMSE), and cross-correlation coefficient (CC).

The Z-R relations for All, convective, and stratiform rain are summarized in Table 2.

**Table 2.** Z-R relations for all, convective, and stratiform rain type at the four studied sites.

| Site | Z-R Relations | | |
| :---: | :---: | :---: | :---: |
| | ALL | Convective | Stratiform |
| BOS | $R = 0.058Z^{0.621}$ | $R = 0.075Z^{0.596}$ | $R = 0.067Z^{0.591}$ |
| BUS | $R = 0.058Z^{0.641}$ | $R = 0.070Z^{0.623}$ | $R = 0.067Z^{0.608}$ |
| CPO | $R = 0.106Z^{0.575}$ | $R = 0.132Z^{0.548}$ | $R = 0.040Z^{0.773}$ |
| JIN | $R = 0.064Z^{0.626}$ | $R = 0.080Z^{0.603}$ | $R = 0.067Z^{0.612}$ |

To compare the characteristics of polarimetric variables with rain types, the occurrence frequency of horizontal reflectivity ($Z_H$), differential reflectivity ($Z_{DR}$), and specific differential phase ($K_{DP}$) were calculated at the four studied sites. Figure 12 shows the occurrence frequency of reflectivity with rain types at each site. The frequency of stratiform rain was larger than that of convective rain weaker than 31 dBZ at BOS, 30 dBZ at BUS, 27 dBZ at CPO, and 29 dBZ at JIN. The average reflectivity of stratiform and convective rain were 22.6 dBZ and 38.0 dBZ at BOS, 22.4 dBZ and 36.6 dBZ at BUS, 21.9 dBZ and 35.1 dBZ at CPO, and 22.4 dBZ and 37.0 dBZ at JIN.

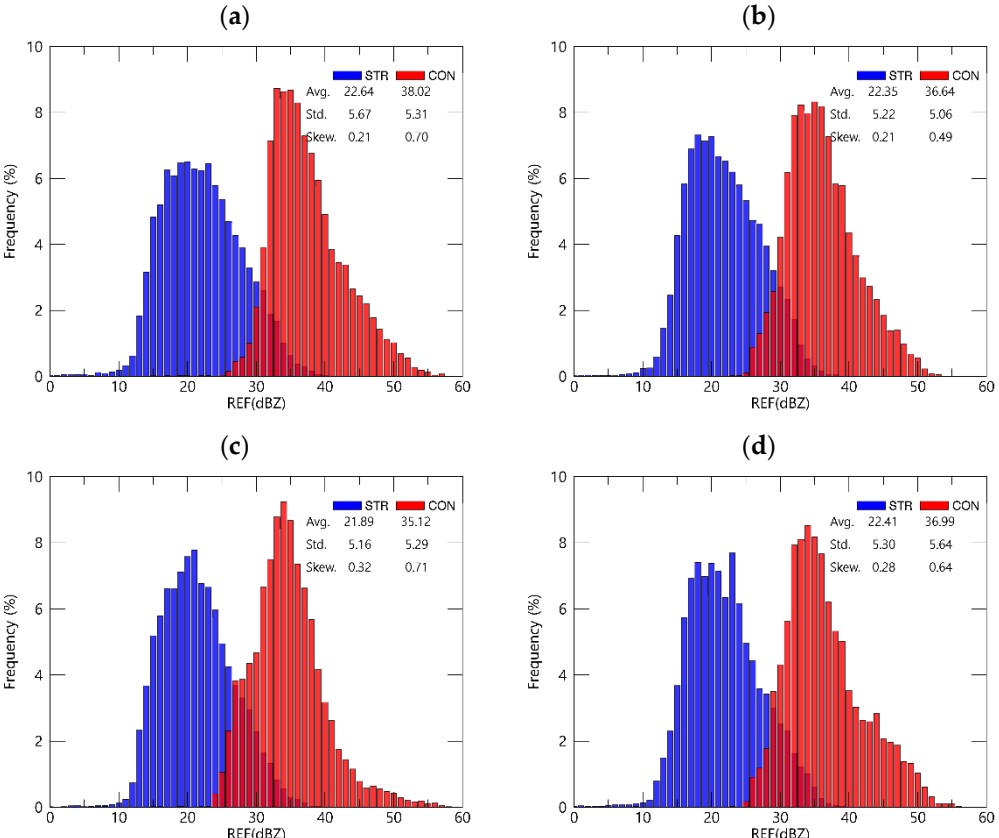

**Figure 12.** The occurrence frequency of reflectivity with rain types at (**a**) BOS, (**b**) BUS, (**c**) CPO, and (**d**) JIN site. The blue represents stratiform rain and the red represents convective rain. The legend shows the average (Avg.), standard deviation (Std.), and skewness (Skew) of reflectivity for both rain types.

Figure 13 shows the occurrence frequency of differential reflectivity with rain types at each site. The frequency of stratiform rain was larger than that of convective rain weaker than 0.4 dB at BOS, 0.3 dB at BUS, 0.3 dB at CPO, and 0.4 dB at JIN. The average differential reflectivity of stratiform and convective rain were 0.24 dB and 0.71 dB at BOS, 0.19 dB and 0.53 dB at BUS, 0.15 dB and 0.48 dB at CPO, and 0.19 dB and 0.57 dB at JIN.

Figure 14 shows the occurrence frequency of specific differential phase with rain types at each site. The frequency of stratiform rain was 100% below 0.1 dB at the four studied sites. The average specific differential phase of stratiform rain for all sites were 0.1 deg km$^{-1}$. The average specific differential phase of convective rain was 0.25 deg km$^{-1}$ at BOS, 0.19 deg km$^{-1}$ at BUS, 0.16 deg km$^{-1}$ at CPO, and 0.22 deg km$^{-1}$ at JIN. This would be one of research topic on separation of rain types using polarimetric radar observation.

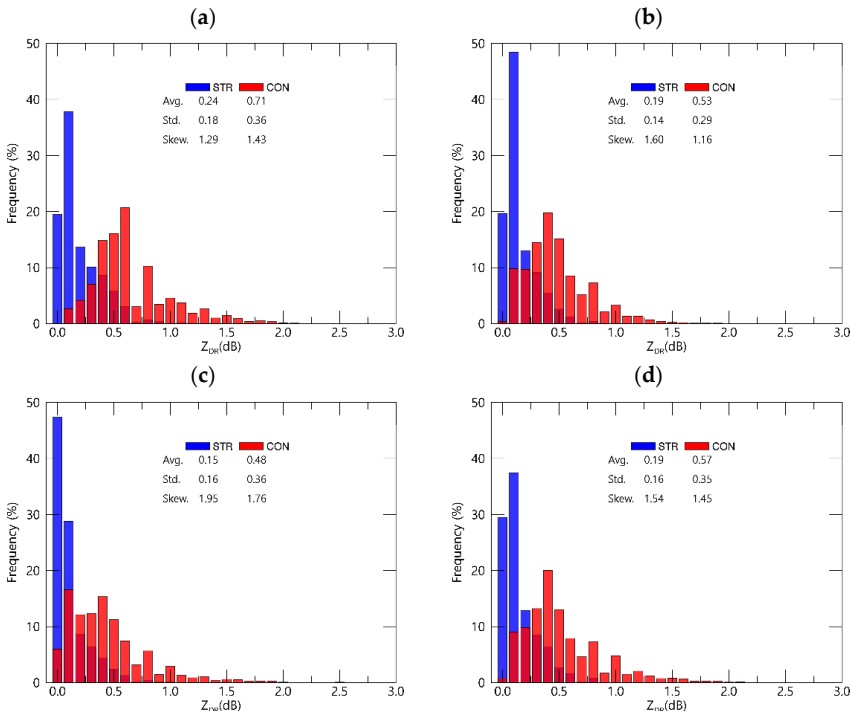

**Figure 13.** The occurrence frequency of differential reflectivity with rain types at (**a**) BOS, (**b**) BUS, (**c**) CPO, and (**d**) JIN site. The blue represents stratiform rain and the red represents convective rain. The legend shows the average (Avg.), standard deviation (Std.), and skewness (Skew) of differential reflectivity for both rain types.

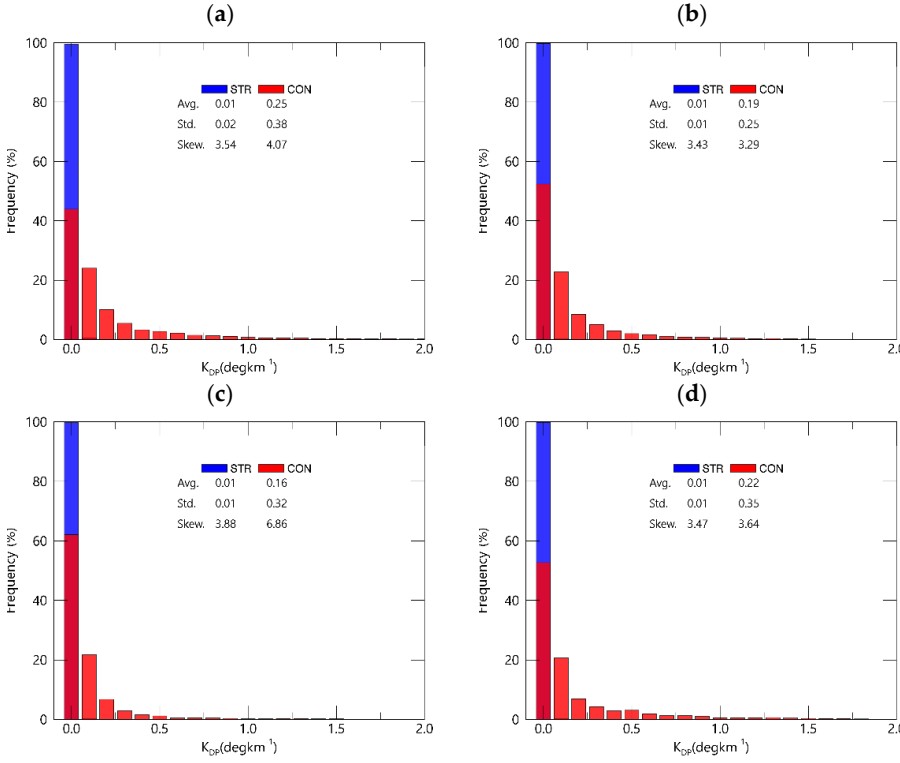

**Figure 14.** The occurrence frequency of specific differential phase with rain types at (**a**) BOS, (**b**) BUS, (**c**) CPO, and (**d**) JIN site. The blue represents stratiform rain and the red represents convective rain. The legend shows the average (Avg.), standard deviation (Std.), and skewness (Skew) of specific differential phase for both rain types.

### 3.4. Raindrop Size Distribution with Wind Direction and Its Hourly Variation

To determine the distributions of the raindrop size with wind direction at the four studied sites, the occurrence frequency of the *Dm* was calculated with the 16 wind directions. This concept comes from the wind rose, which displays the wind speed in different wind directions. In this study, *Dm* was used instead of the wind speed, and it is referred to as the Dm-rose. The program for display of Dm-rose was provided by [48] and modified.

Figure 15 shows the *Dm*-rose with four different categories (0–1 mm, 1–2 mm, 2–3 mm, and over 3 mm) at four different sites. At the BOS site (Figure 15a), the highest proportions of winds were 15.8% in the ESE and 15.7% in the SE, whereas the lowest was 1.1% in the NNE. Small drops of less than 1 mm occurred the most in the ESE, and middle-sized drops between 1 mm and 3 mm appeared more often in the SE than in the other directions. The occurrence frequency of drops larger than 3 mm was 0.1% in the ESE and SSE directions.

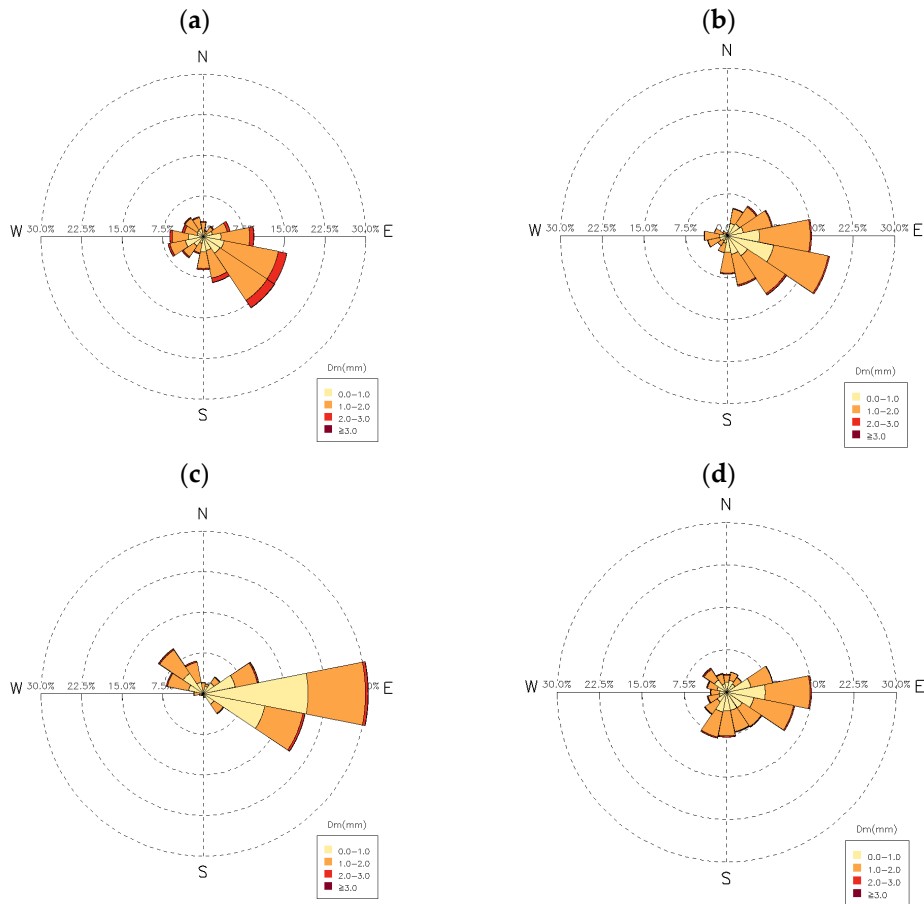

**Figure 15.** The *Dm* distribution with wind direction of the (**a**) BOS, (**b**) BUS, (**c**) CPO, and (**d**) JIN sites.

At the BUS site (Figure 15b), the highest proportions of wind were 18.7% in the ESE and 15.1% in the E, whereas the lowest was 0.6% in the NNW and 0.6% in the N. The small and middle-sized drops (less than 3 mm) occurred the most when the wind direction was ESE, whereas rain drops larger than 3 mm occurred the most in the SE direction. In the case of the CPO site, the east wind direction dominated, with 30.5%, and the second most frequent direction was ESE, with 19.2%. This is clearly different from the other sites, especially for the coastal area of the BOS and BUS sites. All the Dm sizes occurred the most in the E direction and the ratio of the occurrence of small drops (0–1 mm) to middle drops (1–2 mm) was larger than that of the other sites (Figure 15c). At the JIN site (Figure 15d), the east wind direction was also significant, at 15.0%, and the W direction occurred the least often, at 2.9%.

In the coastal areas of BOS and BUS (mountain area of CPO and middle land area of JIN), the dominant wind direction was ESE (E). The Dm-rose is a good indicator, showing the occurrence frequency of different raindrop sizes with the wind direction at a glance.

Figure 16 shows the hourly average *Dm* and *logNw* values at the four studied sites. When Dm is larger, the *logNw* is lower for all sites. The BOS site had the largest *Dm* and the lowest *logNw* for almost every day compared to the other sites. There are four peak values of *Dm* at the BOS site with two peaks in the morning and two peaks in the afternoon. At the BUS site, one peak of *Dm* occurred at 07 LST and one peak was at 18 LST. The lowest *Dm* for BUS occurred at 11 and 12 LST. In the case of the CPO (JIN) site, the peak value of *Dm* was at 08 (16) LST. The *logNw* of CPO (BOS) was larger than that of the other sites. The time variation of the *Dm* was much higher than that of the *logNw*.

(**a**)

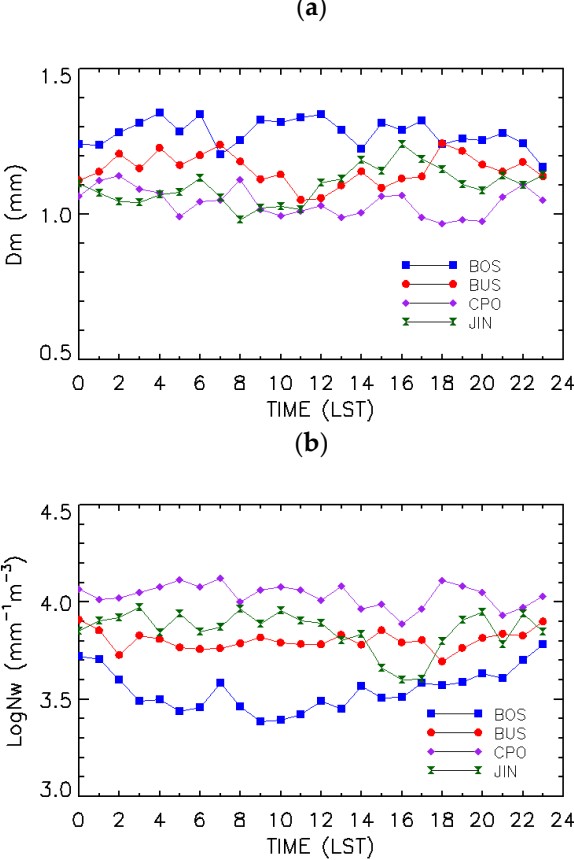

(**b**)

**Figure 16.** Time series of average (**a**) *Dm* and (**b**) *logNw* at the four studied sites. The rectangles in blue represent BOS, the triangles in red represent BUS, the circles in purple represent CPO, and the arrows in green represent the JIN site.

## 4. Conclusions

To understand the microphysical characteristics of rainfall at four different climatological regions (called BOS, BUS, CPO, and JIN) in Korea, DSDs and their variables, including the mass-weighted mean diameter (*Dm*) and normalized number concentration (*logNw*), were examined. We also analyzed the Z-R relations and polarimetric variables at the four studied sites.

In the coastal areas of BOS and BUS, larger raindrops with a lower number concentration mainly comprised the rain, whereas smaller raindrops with a higher number concentration dominated in the mountain region of the CPO and the middle land region of JIN sites. At the CPO site, the number concentration of smaller raindrops (less than 1 mm) was the highest in the weak and moderate rain rate categories. The BOS site had a higher number concentration of diameters larger than 1 mm for the moderate rain rate category. For the severe rain rate category, the trends of the number concentration variation

with diameter were not significantly different among the sites except for BUS, which had a diameter larger than 4 mm.

The contributions of the drop size to the rain rate and total number concentration were analyzed. The mountain area of CPO and middle land area of JIN had a larger contribution to the rain rate than that of the coastal area of BOS and JIN in the range of the smallest diameter. For the larger raindrop diameter, the pattern was the reverse. The contribution of the drop size to the total number concentration at the CPO and JIN sites was larger (smaller) than that at BOS and BUS for the smallest (larger) diameter.

The features of the DSD variables with the rain types, stratiform and convective rain, were analyzed. The proportion of stratiform rain was higher in the mountain and middle land area than that in the coastal area. This caused a larger (smaller) Dm and lower (higher) logNw at the BOS (JIN) and BUS CPO) sites for both rain types. The average shape and slope parameter of gamma model were higher values at the mountain region of CPO site than at other sites for both rain types.

The intercept coefficient of Z-R relation showed higher values in the mountain area and middle land area than the coastal area, whereas the slope coefficient had the smallest values in the mountain area. The polarimetric variables of $Z_H$ and $Z_{DR}$ were shown highest (lowest) value at the coastal region of BOS (mountain area of CPO) site for both rain types. These results are consistent with larger Dm at the coastal area than that at the mountain area. The average $K_{DP}$ had same value at the four studied sites for stratiform rain and had the highest (lowest) value at the coastal area of BOS (mountain area of CPO) for convective rain.

The *Dm*-rose, which shows the Dm distributions with wind direction, was proposed in this study. In the coastal area (mountain and middle land area), the dominant wind was ESE (E). The ratio of the smaller diameter to the middle size at BOS was much smaller than that at CPO. The *Dm*-rose is a good indicator of the occurrence frequency of different raindrop sizes with wind direction. In the analysis of the hourly distribution of the *Dm* and *logNw*, there were two and four peaks of the Dm at BUS and BOS, respectively, whereas there was one peak of the *Dm* at the CPO and JIN sites. The time variation of the *Dm* is much higher than that of the *logNw*.

Even though the DSDs of only summer seasons of three years were analyzed, we found that different characteristics of DSDs, Z-R relations, and polarimetric variables occurred in the different climatological regions of South Korea in this study. These results can contribute to further research on weather radar rainfall estimates and rain type classification.

**Author Contributions:** Conceptualization, C.-H.Y. and M.-Y.K.; methodology, M.-Y.K.; writing—original draft preparation, C.-H.Y.; writing—review and editing, C.-H.Y., M.-Y.K., H.-J.K., S.-H.S. and W.J.; visualization, C.-H.Y., H.-J.K. and S.-H.S.; project administration, C.-H.Y.; funding acquisition, C.-H.Y. All authors have read and agreed to the published version of the manuscript.

**Funding:** This research was supported by Basic Science Research Program through the National Research Foundation of Korea (NRF) funded by the Ministry of Education (NRF-2020R1I1A3066504).

**Institutional Review Board Statement:** Not applicable.

**Informed Consent Statement:** Not applicable.

**Data Availability Statement:** Not applicable.

**Conflicts of Interest:** The authors declare no conflict of interest.

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
