# Peer review of "The Characteristics of Raindrop Size Distributions in Different Climatological Regions in South Korea"

_remotesensing, doi:10.3390/rs14205137_

Round 1
Reviewer 1 Report (Previous Reviewer 2)
Reviewer
MDPI Remote Sensing
Manuscript Number: ID 1956890 /1893442-V3
Title: The characteristics of raindrop size distributions is different climatological regions in South Korea
As requested, I have reviewed the second revised version (ID 1893442-V3) of the above-titled paper for potential publication in the Remote Sensing- MDPI Journal, presently submitted as manuscript ID 1956890. I divided my comments in the sections presented as follows.
Contribution
This paper proposes the investigation of microphysical characteristics of precipitation at four different climatological regions in South Korea (Figure 1): Daegwallyeong (CPO), Jincheon (JIN), Boseong (BOS) and Busan (BUS). As the authors mentioned,, two of them are close to the coast and lower altitude (BOS and BUS), while the other two are inland (CPO – high latitude and higher altitude; JIN – middle latitude and lower altitude. More specifically, the research work focused on raindrop size distributions (DSDs) and corresponding variables, notably mean diameter (Dm) and normalized number concentration (logNw). Parsivel disdrometers have been used during the rainy seasons from June to August for three (3) years (2018-2020). The authors explored the better understanding of stratiform and convective rainfall for these four sites. The authors also explored relationships of Dm with respect to wind direction among other analysis explored in the manuscript sand summarize their conclusions with respect to their findings.
In this revised version, the authors included new and important results related to weather radar datasets including the analysis of Z-R relationships for stratiform and convective systems as has been pointed out in the first revision of the manuscript 1893442-V1) but that have been left out of the analysis by the authors in the second revised version of the manuscript (1893442-V2)
There was definitely an improvement of the manuscript in terms of its contents. I found the manuscript has an interesting goal to be pursued and presents the conditions to be published in the MDPI – Remote Sensing Journal.
However, the authors should presently revise the whole text of the manuscript to take into account the new results including weather radar information (abstract, introductory section, datasets, methodology). In particular, mistypes, grammar correction, inclusion of figure, renumbering of figures and proper legends for figures are required for adequate editing of whole text of the manuscript.
Therefore, brief comments are made in the next section to provide the authors guidance and support to conduct a thorough review of the manuscript based on this new version.
Please, see further comments in the attached file.

Author Response
Please refer to the attached file.

Reviewer 2 Report (Previous Reviewer 3)
General comments:
This paper analyzed the microphysical characteristics of raindrops at four stations during the rainy season in South Korea based on Parsivel observations from 2018 to 2020. The regional comparison showed that the contribution of raindrops of different sizes to total number concentration and rain intensity was also significantly different. The authors explored a better understanding of stratiform and convective rainfall for these four sites in south Korea. The Dm-Rose graph first presented by the authors showed some interesting results. Nevertheless, empirical relationships were also well demonstrated. In general, the authors have used valuable observations over a period of three years from four different locations to advance our understanding of rain microphysical processes and features in South Korea. A revision is needed before it can be accepted for publication. Here are the comments below:
Major comments:
1. What was the radar wavelength used in T-matrix scattering simulation? Besides, the workflow in Fig 3. didn’t describe the T-matrix scattering simulation, which just including part of the work presenting in this paper. The proper distribution and completeness of workflow can be rethought.
2. As shown in the section 3.2, the authors have divided different rain types. It is regrettable that the comparisons between rain rate measured by DSD and R(Z) relations were presented just from four sites in the section 3.3. Detailed comparisons under total, convective and stratiform rain types is recommended, which may improve the accuracy of the R(Z) relationship and be helpful to subsequent analysis.
3. From Fig. 1, we find that the altitude of CPO site is close to 1000 m, while the other are below 150 m. Although the authors have been aware of the effect of terrain height on fall velocity, it’s better to multiply a correction factor to take the terrain height of the observation site into account instead of analyzing it in future studies. The correction factor proposed by Atlas et al. (1973) can be referred to.
Minor comments:
1. Line 27: Replace ‘.’ with ‘,’.
2. The authors used ‘Dm-rose’ in Abstract, Keywords and Line 553, while ‘Dm rose’ were used in Line 134, 249, 493, 474, 501, 556. Also. the font is not uniform. Authors should use the same representation throughout the manuscript.
3. The authors should use the italic font for parameters with physical meaning (e.g., Z should be Z etc.).
4. Line 241-242:The sentence ‘Other … temperature.’ is incomplete. Please modify it.
5. Fig.3: The formula at the bottom left of the ‘Quality control’ is wrong. Please check it.
6. Replace the horizontal coordinates of Fig. 8(9) with μ(Λ(mm-1)).
7. Line 390: The font size of the first ‘mm-1’ is different from other ‘mm-1’. Please modify it.
8. Throughout the manuscript, the form of the parameters in the pictures and the text needs to be the same (e.g., ZDR in Fig.12 should be ZDR etc.).
9. Line456: Replace ‘degkm-1’ with ‘degkm-1’.
Reference:
Atlas, D., Srivastava, R. C., and Sekhon, R. S. (1973), Doppler radar characteristics of precipitation at vertical incidence, Rev. Geophys. 11(1), 1–35.
Round 2
Reviewer 1 Report (Previous Reviewer 2)
Reviewer
MDPI – Remote Sensing
Manuscript Number: ID 1956890-v2
Title: The characteristics of raindrop size distributions in different climatological regions in South Korea
As requested, I have reviewed the revised version of the above-titled paper for potential publication in the Remote Sensing - MDPI Journal. I divided my comments in the sections presented as follows.
Contribution
This manuscript proposes the investigation of microphysical characteristics of precipitation at four different climatological regions in South Korea. There was significative improvement since the first draft of the manuscript.
I found the manuscript has an interesting goal to be pursued and presents the conditions to be published in the MDPI Remote Sensing Journal.
Please, see further comments in the attached file.

This manuscript is a resubmission of an earlier submission. The following is a list of the peer review reports and author responses from that submission.
Round 1
Reviewer 1 Report
In this manuscript the authors tried to study the RSD characteristics from four different locations in South Korea. The findings in the present manuscript were already reported for the South Korea region. Moreover, the present manuscript doesn’t provide any new results, which are crucial for the South Korea region. Hence, the present version of the manuscript may not be suitable for the publication in remote sensing journal. The authors need to provide new and more findings in the manuscript. The authors can consider the below mentioned points while revising/updating the manuscript.
1. In the third paragraph of the introduction section (lines 42-51), the authors tried to mention about the polarimetric radar. As the present manuscript didn’t use any radar, I wonder how this paragraph is appropriate for this study.
2. Page2, Lines 59-63, 82-98: In this paragraph, authors discussed about RSD studies from Italy, India, and South Africa. It would be more appropriate to detail about the RSD studies from the neighbouring/nearby countries/locations like, Japan, Taiwan etc., rather than talking about the RSD studies from the far away countries. Also, authors should provide more emphasis on the previous RSD studies over South Korea, like what was reported by the previous researchers, what is lack in the literature especially related to RSD, etc.
3. Over all, the introduction section lacks the continuity and major reasons/objectives for the present study.
4. The findings presented in this study are similar to the previous studies over south Korea. There are no news findings in the present manuscript. Hence, the authors can refer to the previous RSD studies over South Korea to improve the standards of the present manuscript.
5. The authors didn’t provided pros and cons of the Parsivel disdrometer, and the quality control procedure adapted to the disdrometer raw data.
6. The results provided in this manuscript are weak. It is strongly recommended to provide more results, and those results should provide superior information than the existing literature for South Korea.
Author Response
Dear Reviewer,
Authors attached the answer to the reviewer's comments.

Reviewer 2 Report
Reviewer
MDPI Remote Sensing
Manuscript Number: ID 1893442
Title: The characteristics of raindrop size distributions is different climatological regions in South Korea
As requested, I have reviewed the above-titled paper for potential publication in the Remote Sensing- MDPI Journal. I divided my comments in the sections presented as follows.
Contribution
This paper proposes the investigation of microphysical characteristics of precipitation at four different climatological regions in South Korea (Figure 1): Daegwallyeong (CPO), Jincheon (JIN), Boseong (BOS) and Busan (BUS). As the authors mentioned,, two of them are close to the coast and lower altitude (BOS and BUS), while the other two are inland (CPO – high latitude and higher altitude; JIN – middle latitude and lower altitude. More specifically, the research work focused on raindrop size distributions (DSDs) and corresponding variables, notably mean diameter (Dm) and normalized number concentration (logNw). Parsivel disdrometers have been used during the rainy seasons from June to August for three (3) years (2018-2020). The authors explored the better understanding of stratiform and convective rainfall for these four sites. The authors also explored relationships of Dm with respect to wind direction among other analysis explored in the manuscript sand summarize their conclusions with respect to their findings.
I found the manuscript has an interesting goal to be pursued and presents the conditions to be published in the MDPI – Remote Sensing Journal. However, the text needs to be fully revised.
In general, the basic ideas, concepts and assumptions are presented along the manuscript but not necessarily in the best order. The manuscript is not clear enough along the development of some of the steps proposed in the methodological approach and with respect to some of the results presented. There are some missing links and the manuscript does not present a clear work flow, even though the authors describe most of the procedures.
Taking that into account, there are further questions and doubts I would like to hear from the authors. The reader needs to have those points better clarified. Figures and Tables can be improved and possibly complemented to adequately present results. That might also lead to explore or reflect about different scenarios still not well and thoroughly explored by the authors in the proposed paper but that deserves attention.
Please, see further comments in the attached file.

Author Response
Dear Reviewer,
Attach is the answer to the reviewer's comments.

Reviewer 3 Report
General comments:
This paper analyzed the microphysical characteristics of raindrops at four stations during the rainy season in South Korea based on Parsivel observations from 2018 to 2020. DSDs and microphysical parameters were examined, including the mass-weighted mean diameter (Dm) and normalized number concentration (Nw). The Dm-Rose Diagram showed some interesting results. In addition, convective and stratiform rain characteristics were studied based on the rain rate classification methodology. The regional comparison showed that the contribution of raindrops of different sizes to total number concentration and rain intensity was also significantly different. But overall, compared with previous studies, this paper did not provide new scientific perspectives to advance our understanding of rain microphysical processes and features in South Korea.
Major Comments:
1. As the authors indicate, there have been many studies that have looked at the precipitation microphysics in South Korea by using different disdrometers. The climatological characteristics of raindrop size distributions have been well documented (You and Lee, 2015; Suh et al., 2016; Cha and Yum., 2021). This manuscript does the same thing, but what is new in what is shown here? I realize that this study has four disdrometers that may be able to reveal the temporal and spatial variations of DSDs, which could help improve the accuracy of quantitative precipitation estimates and forecasts. I suggest the author should find a point that is different from previous findings.
2. More evidence is needed on the representation of the four sites for different climate regions, especially continental climates. First, from Figure 1, we find that the distance of the four stations from the coast is no more than 100 km; Second, Figure 6 shows the distribution of Dm and log10Nw average values of convective and stratiform rain for each site. No data points fall within the continental-like region defined by Bringi (2003) (<Dm >~2 -- 2.75 mm and log10< Nw >~3 -- 3.5), which may require further discussion.
3. The separation of convective and stratiform precipitation introduced in section 2 is questionable. The classification scheme of Bringi et al. (2003) was based on the standard deviation of rainfall rate of five consecutive 2-minute DSD samples, while the authors used the standard deviation of five 1-minute samples, which may affect the precipitation classification results because of the length of the continuous DSD samples.
Minor comments:
1. Please unify the expression of microphysical parameters in this paper.
2. Line 14. Substitute “in” for “at”.
3. Line 20-22. Improve wording to clarify the meaning of the sentences.
4. Line 24. This sentence is confusing. Please improve wording.
5. Line 135. Fig. 1. The locations of four disdrometers are not easily distinguishable. Using other color schemes would be recommendable.
6. Line 154. The Eqs (3) is wrong, please check it.
7. Line 197-199. This sentence is ambiguous. Please modify it.
8. Line 197-199. This sentence is ambiguous. Please modify it.
9. Line 269. Table 1. Units need to be supplemented.
10. Line 323-325. Adding a physical analysis to this paragraph would be better.
11. Line 330. What does SLT stand for? Substitute “LST” for “SLT”.
12. Line 343-348. Fig 8. What is the purpose of presenting this figure?
Author Response

(The authors gave the same response as above.)

Round 2
Reviewer 1 Report
The authors tried to improve the manuscript by considering previous comments. By looking at the revision version of the manuscript, I noticed few more points/correction that are needed to improve the quality of the manuscript. Hence, the authors can consider the below mentioned points while revising the manuscript.
Minor comments:
1. Throughout the manuscript, the authors should use the italic font for RSD parameters (Ex: Dm should be Dm etc.).
2. Page 1, Abstract: In the abstract, the authors should take care mention the subscripts to the RSD parameters (for example Dm should be Dm). Also throughout the manuscript, the authors should do the necessary modifications.
3. Page 1, line 28: expand “ESE (E)”.
4. Page 4-5: Figure 2 subplots should be drawn in in one figure window, and Figure 2 should appear in one page only.
5. Page 2, line 129: At line 129, the authors used “Dm-ROSE”, but in the Abstract and at line # 218 it is shown as “Dm rose”. The authors should use same notation/representation throughout the manuscript.
6. Page 7-8, Figure 5: The subplot should be kept in one page. I suggest to keep three subplots in one row and three column fashion.
7. Page 9, line 319: “LogNw” should be “logNw”
8. Page 10-11, Figure 9: All the subplots of Figure 9 should be placed in one page.
9. Page 11-12: Please check the figure number (I think it should be Fig. 10) for the figure given in page 11-12.
10. The subplots of the figure provided in Page 11-12 should be kept in one page. It is recommended to modify the subplots orientation similar to figure 6.
Author Response
Please refer to the attach.

Reviewer 2 Report
Reviewer
MDPI Remote Sensing
Manuscript Number: ID 1893442-V2
Title: The characteristics of raindrop size distributions is different climatological regions in South Korea
As requested, I have reviewed the revised version of the above-titled paper for potential publication in the Remote Sensing- MDPI Journal. I divided my comments in the sections presented as follows.
Contribution
This paper proposes the investigation of microphysical characteristics of precipitation at four different climatological regions in South Korea (Figure 1): Daegwallyeong (CPO), Jincheon (JIN), Boseong (BOS) and Busan (BUS). As the authors mentioned,, two of them are close to the coast and lower altitude (BOS and BUS), while the other two are inland (CPO – high latitude and higher altitude; JIN – middle latitude and lower altitude. More specifically, the research work focused on raindrop size distributions (DSDs) and corresponding variables, notably mean diameter (Dm) and normalized number concentration (logNw). Parsivel disdrometers have been used during the rainy seasons from June to August for three (3) years (2018-2020). The authors explored the better understanding of stratiform and convective rainfall for these four sites. The authors also explored relationships of Dm with respect to wind direction among other analysis explored in the manuscript sand summarize their conclusions with respect to their findings.
I found the manuscript has an interesting goal to be pursued and presents the conditions to be published in the MDPI – Remote Sensing Journal.
Please, see further comments in the attached file.

Author Response
Please refer to the attach.

Reviewer 3 Report
General comments:
This paper analyzed the microphysical characteristics of raindrops at four stations during the rainy season in South Korea based on Parsivel observations from 2018 to 2020. DSDs and microphysical parameters were examined, including the mass-weighted mean diameter (Dm) and normalized number concentration (Nw). The Dm-Rose graph showed some interesting results. In addition, convective and stratiform rain characteristics were studied based on the rain rate classification methodology. The regional comparison showed that the contribution of raindrops of different sizes to total number concentration and rain intensity was also significantly different.
The authors made corresponding changes at the suggestion of the reviewers, and pointed out that some of the work was presented for the first time. However, I believe that more in-depth and enlightening results should be obtained using raindrop spectra from four regions in South Korea over three years.
In view of this, the manuscript may not suitable for publication in Remote Sensing journal at this time, but resubmission is encouraged after modifications and additions to the following.
1. In the introduction section, the review of previous studies should focus on the scientific objectives of this research, such as precipitation climate characteristics of latitudes or adjacent regions, microphysical characteristics of raindrop spectrum, etc.
2. Although the authors mention that the results of this study are the first to be obtained using a dataset of long-term raindrop spectra from four different locations, the overall analysis results are not in-depth enough to reflect the strengths of this dataset. More research should be carried out, such as gamma function fitting coefficient, Z-R relationship, even dual-polarization radar retrieval, etc.
3. The authors added the flow description of raindrop spectrum processing and specified the climatic region names of the four observation sites. According to the modified Figure 1, the altitude of CPO is close to 1000 meters, but the influence of altitude on the falling speed of raindrops is not mentioned in the paper, which should be corrected.
4. The separation of convection and stratified precipitation presented in Section 2 needs to be discussed. An evaluation of the classification results by a new method should be given to convince readers of the method proposed by the authors. The precipitation classification method proposed by Chen et al. (2013) based on Bringi et al. (2003) and Marzano et al. (2010) has been adopted in most literature. The authors can compare the difference in classification results between Chen's method and the method in this paper.
5. The DM-Rose graph as a highlight would appear better if it was accompanied by a weather situation or microphysical interpretation to clarify its application.
Reference
Bringi, V. N., V. Chandrasekar, J. Hubbert, E. Gorgucci, W. L. Randeu, and M. Schoenhuber, 2003: Raindrop Size Distribution in Different Climatic Regimes from Disdrometer and Dual-Polarized Radar Analysis. Journal of the Atmospheric Sciences, 60, 354-365.
Chen, B. J., J. Yang, and J. P. Pu, 2013: Statistical Characteristics of Raindrop Size Distribution in the Meiyu Season Observed in Eastern China. Journal of the Meteorological Society of Japan, 91, 215-227.
Marzano, F. S., D. Cimini, and M. Montopoli, 2010: Investigating precipitation microphysics using ground-based microwave remote sensors and disdrometer data. Atmospheric Research, 97, 583-600.
Author Response
Dear reviewer,
We tried to answer and modified the manuscript according to the comments. However, there are some comments out of the scope of this paper. Please understand these and refer to the attach.
